# Analysis of the Role of Gene Variants in Matrix Metalloproteinases and Their Tissue Inhibitors in Bronchopulmonary Dysplasia (BPD): A Study in the Polish Population

**DOI:** 10.3390/cimb48010025

**Published:** 2025-12-25

**Authors:** Dawid Szpecht, Kareem Abu-Amara, Grażyna Kurzawinska, Agnieszka Seremak-Mrozikiewicz

**Affiliations:** 1Department of Neonatology, Poznan University of Medical Sciences, 60-535 Poznan, Poland; 2Poznan University of Medical Sciences, 60-512 Poznan, Poland; kareemamara599@gmail.com; 3Department of Perinatology, Poznan University of Medical Sciences, 60-535 Poznan, Poland; gene@gpsk.ump.edu.pl (G.K.); asm@data.pl (A.S.-M.)

**Keywords:** prematurity, bronchopulmonary dysplasia, matrix metalloproteinases, MMP-1, MMP-9, tissue inhibitors of matrix metalloproteinases

## Abstract

This study aimed to explore the association between genetic variants of matrix metalloproteinases (MMP-1 rs1799750, MMP-9 rs17576, and rs17577) and their tissue inhibitors (TIMP-1 rs4898, TIMP-2 rs2277698, and rs55743137) in the development of bronchopulmonary dysplasia (BPD) in infants from a Polish population. Methods: A cohort consisting of 100 premature infants (47% female) was analyzed, in which there were 38 BPD cases and 62 controls without BPD. Genotype distributions were analyzed, and their relationship with BPD risk was assessed after adjustment for potential confounders. Results: Application of Bonferroni correction for multiple testing showed that none of the single-nucleotide polymorphisms (SNPs) reached the adjusted significance threshold (*p* < 0.008). However, analysis of allele frequencies using adjusted *p*-values identified a statistically significant difference for MMP1 rs17999750 (*p* = 0.038). Conclusion: These findings do not support a significant role of TIMP-2 and MMP-9 genetic variations in the pathogenesis of BPD among preterm infants. While these results are informative, a limitation of this study is the small sample size, and larger studies are needed to confirm these observations.

## 1. Introduction

First used by Northway et al. in 1967, the term “bronchopulmonary dysplasia (BPD)” describes the chronic form of injury towards the lungs due to barotrauma and oxygen injury in infants requiring mechanical ventilation [1]. In 1999, Jobe introduced the term “new BPD” to describe the form of chronic lung disease seen in preterm infants at that time. Unlike the “old BPD,” which involved abnormal microvasculature and simplified alveoli, the “new BPD” was marked by less airway injury and reduced alveolar septal fibrosis [2].

Despite significant advancements in the field, such as antenatal steroids and surfactant therapy, the incidence of BPD has remained unchanged and continues to be the most frequent adverse outcome of prematurity [3]. This can be attributed to the aforementioned advancements allowing for the survival of premature infants that present with lower gestational ages (GAs) and the survival of very low birth weights (VLBWs), as these have resulted in a change in the characteristics of BPD [4]. The same notion applies to primary prevention; although promising research has been conducted, a consistently effective strategy for preventing bronchopulmonary dysplasia has yet to be achieved. In the United States, approximately 10,000 preterm infants are estimated to be afflicted by BPD annually [5]. The global rate of BPD among infants born extremely premature (<28 weeks GA) ranges from 10 to 89%, with rates of 10–73% in Europe and 18–89% in North America [6]. As of the fiscal year 2023, Poland’s birth rate was 273,000, with 7.2% being preterm [7]. One study on a Polish population found that 52.7% of infants born before 32 weeks and weighing ≤1500 g developed BPD [8]. Another study, involving extremely preterm infants <28 weeks in Poland, reported a higher BPD rate of 80.9%, with most cases classified as mild [9]. Thus, Poland’s BPD rates are within the range reported for Europe and North America.

The scientific literature shows that BPD has a multifactorial pathology, influenced by a variety of pre- and postnatal factors affecting both the mother and infant. Some of these include pre-eclampsia, maternal smoking, maternal and fetal infection, and genetic susceptibility, the latter being the focus of our study [10]. Matrix metalloproteinases (MMPs) belong to a family of zinc-dependent endopeptidases, also known as matrixins. They play a crucial role in the degradation and remodeling of the extracellular matrix. They cleave various extracellular components and cell surface-associated proteins and also target intracellular substrates. MMPs also degrade major components such as collagen, elastin, and gelatin. The degradation of these components plays a vital role in various physiological processes, including embryonic development, angiogenesis, tissue formation, reproduction, and the resorption and remodeling of tissues [11,12]. Alterations in MMP expression can generate abnormal breakdown of the extracellular matrix (ECM).

Tissue inhibitors of MMPs (TIMPs) regulate MMP activity within the ECM by blocking their proteolytic effects. TIMPs play an important role as regulators of ECM turnover, tissue remodeling, and cellular behavior. Similar to MMPs, they influence processes such as angiogenesis, cell growth, and apoptosis. Disruption of the balance between MMPs and TIMPs has been linked to the pathogenesis and progression of various diseases [13].

Based on current knowledge, elevated MMP activity is linked to abnormal ECM remodeling in the lungs, contributing to the development of BPD. One study, conducted in a Japanese cohort, demonstrated that an elevation in MMP-9/TIMP-1 ratios in cord blood of 29 premature infants who were <30 weeks gestation was linked to an increased risk of moderate to severe BPD. Such imbalances were correlated with prolonged oxygen therapy requirements in preterm neonates [14]. On top of that, low TIMP-2 serum levels at birth have been identified as predictors of BPD development. This deficiency may play a role in the development of BPD in preterm infants by contributing to early lung inflammation [15].

This study aims to thoroughly examine the relationship between variants of MMP-1, MMP-9, TIMP-1, and TIMP-2 genes and BPD in premature infants, with analyses including single-variant tests of association, multi-locus models, and haplotype analysis. Understanding how variation in these candidate genes contributes to the onset and progression of BPD may offer new insights into the underlying mechanisms of the condition

## 2. Materials and Methods

The study was carried out in compliance with the Declaration of Helsinki and received approval from the Bioethics Committee of Poznan University of Medical Sciences (45/22 and 126/22, applied to both control and case groups).

### 2.1. Study Population

This prospective study was carried out at the Clinical Hospital of Gynecology and Obstetrics, Poznan University of Medical Sciences, Poznan, Poland, between 1 March 2014, and 14 January 2020. The study population was made up of 100 consecutive live preterm infants. (1) Preterm delivery between 22 + 0 and 33 + 0 weeks of GA and (2) signed parental/guardian consent for the infant’s participation were the prerequisites for eligibility in the study. Based on clinical and respiratory evaluations, the study population was divided into two groups: the cases (preterm infants diagnosed with BPD; *n* = 38) and the controls (preterm infants without BPD; *n* = 72). All infants and their parents were of Caucasian origin.

Exclusion criteria for both the case and control groups included chromosomal abnormalities, multiple pregnancies, pregnancies complicated by the loss of one fetus, death prior to reaching 40 weeks postmenstrual age, and confirmed inherited metabolic disorders. Patients with congenital infections such as toxoplasmosis, rubella, cytomegalovirus, herpes, or other TORCH pathogens were excluded to eliminate potential confounding effects of systemic infection and inflammation on BPD development. None of the mothers had pre-eclampsia or were smokers.

### 2.2. Clinical Features

Clinical characteristics potentially affiliated with BPD development were documented. These included sex, BW (grams), GA (weeks), mode of delivery, and Apgar scores at 1 and 5 min. Additional recorded factors were duration of ventilatory support (days), birth asphyxia (defined as an Apgar score < 6 at 10 min and either pH < 7.0 or cord blood base excess [BE] < −15 mmol/L), intrauterine infection (confirmed by a positive culture from originally sterile sites along with clinical symptoms or pneumonia within the first 48 h), intrauterine infection (including pneumonia, sepsis, or urinary tract infection), and prematurity-related complications such as ROP, IVH, and NEC.

### 2.3. BPD Diagnosis

The diagnosis of BPD is made in a clinical setting. This diagnosis takes into account the GA, postmenstrual age (PMA), duration of oxygen therapy, and the oxygen requirement at 36 weeks PMA. The National Institutes of Health (NIH) consensus definition (2001) proposed a definition where infants born at less than or equal to 32 weeks GA who require supplemental oxygen for at least 28 days are diagnosed with BPD. The severity of the disease can be classified as mild, moderate, or severe and is determined at 36 weeks PMA based on the infant’s respiratory support status at that time [16].

The updated evidence-based classification proposed in 2019 defines BPD severity based on the type of respiratory support needed at 36 weeks PMA [17]. This classification was not applied to the current dataset due to its retrospective design and the absence of detailed information on respiratory support

### 2.4. Studied Genetic Variants

Targeted genes were chosen for this study on the premise of their role within the development of BPD. This study chose target genes that were chosen based on their potential role in the development of BPD. Single-nucleotide variants were obtained from the National Center for Biotechnology Information (NCBI) dbSNP database (http://www.ncbi.nlm.nih.gov/projects/SNP, accessed on 10 January 2023). The variants showed a minor allele frequency (MAF) of at least 5% within European populations, and these variants consisted of MMP-1 rs179975, MMP-9 rs17576, MMP-9 rs17577, TIMP-1 rs4898, TIMP-2 rs2277698, and TIMP-2 rs55743137. Table 1 presents details of these variants.

Peripheral venous blood samples of 0.5 mL were collected from participants and stored for analysis. DNA extraction was carried out using the QIAamp DNA Blood Mini Kit (QIAGEN Inc., Hilden, Germany) in accordance with the manufacturer’s protocol. Genotyping of the targeted polymorphisms was carried out with the use of polymerase chain reaction (PCR) alongside the restriction fragment length polymorphism (RFLP) technique. Primer sequences and their restriction enzymes (Thermo Fisher Scientific, Waltham, MA, USA) were selected based on previously published studies and are detailed in Table 2 [18,19,20,21,22]. Amplified products were separated by agarose gel electrophoresis and visualized with Midori Green Advance DNA Stain (Nippon Genetics, Düren, Germany). To confirm genotyping accuracy, a subset of samples (approximately 5%) was reprocessed in a blinded fashion. All analyzed variants met quality thresholds, with call rates exceeding 95%. 

### 2.5. Statistical Analysis

The data were summarized using frequencies and percentages for categorical variables and medians with interquartile ranges for continuous variables that were not normally distributed. The Shapiro–Wilk test was used to assess normality. Associations between categorical variables and BPD were evaluated using the Fisher exact probability test, the χ^2^ test, the Fisher–Freeman–Halton test, and the χ^2^ test with Yates correction. For non-normally distributed continuous variables, group differences were analyzed with the Mann–Whitney test.

The links between the genetic variants and BPD were first estimated with univariate analyses (odds ratio, OR) and then with multivariate analyses (adjusted odds ratio, AOR) that controlled for gestational age and birth weight. The corresponding 95% confidence intervals (95% CI) were derived from logistic regression. The variants were checked for conformity to Hardy–Weinberg equilibrium (HWE). To determine how each variant relates to BPD, multiple genetic inheritance models (codominant, dominant, recessive, overdominant, and log-additive) were tested, with the optimal model being selected via the Akaike information criterion (AIC). *p*-values less than 0.008 were considered statistically significant after applying Bonferroni correction for the six SNPs analyzed.

Linkage disequilibrium (LD) among the variants was calculated with Haploview v.4.2 software. A *p*-value of less than 0.05 was considered statistically significant. All statistical analyses were performed using R software version 4.5.0 and the SNPassoc package [23,24,25]

We computed the statistical power under the assumption that 52.7% of preterm infants in the Polish population had BPD [8]. In our cohort, 38% of preterm infants had BPD. With a sample size of N = 100 and alpha = 0.05, the power of the test was 84.4%.

## 3. Results

### 3.1. Clinical Data

The studied infants’ clinical and demographic characteristics are outlined in Table 3. A cohort of 100 infants (47 female) was included in the study and consisted of 62 infants without BPD and 38 infants diagnosed with BPD. The median GA was 26 weeks (range 22–31 weeks), and the median body weight (BW) was 883 g (range 432–1700 g). As shown in Table 3, a lower GA and BW presented a significant association with the development of BPD. Furthermore, it was indicated that infants that developed BPD had significantly lower Apgar scores at both the 1st and 5th minute after birth; they also required a markedly greater period of mechanical ventilation (median 43 days vs. 7 days).

The analysis of the BPD group revealed a significantly increased frequency of prematurity-related complications, namely late-onset infection, Intraventricular Hemorrhage (IVH), and Necrotizing Enterocolitis (NEC). Retinopathy of Prematurity (ROP) was found to have a strong association with BPD, with 76.3% of infants with BPD being diagnosed with ROP compared to only 33.9% in the no-BPD group. There were no statistically significant differences between the groups in regard to incidence of intrauterine infection, birth asphyxia, mode of delivery, or sex distribution.

### 3.2. Association Studies

#### 3.2.1. Single-Variant Tests of Association

Table 4 and Table 5 show the genotype and allele distributions for the investigated MMP-1, MMP-9, TIMP-1, and TIMP-2 variants in infants with and without BPD.

Using the adjusted *p*-value, a statistically significant difference was observed for MMP1 rs17999750 (*p* = 0.038, Table 4).

With the use of multiple genetic models, we were able to further investigate the association between each genetic variant and BPD susceptibility. The results of these analyses, including both the crude odds ratios (OR) and adjusted odds ratios (AOR) for GA and GW, are detailed in Table 5. For the X-linked TIMP-1 rs4898 variant, analyses were performed in males and females separately. The codominant genetic model was used as the main model; in addition, results were obtained for dominant, recessive, overdominant, and log-additive models (Table 5)

After applying Bonferroni correction for multiple testing, none of the single-nucleotide polymorphisms (SNPs) reached statistical significance. The adjusted significance threshold was *p* < 0.008, and all *p*-values were above this cutoff, indicating that no associations remained significant after correction for multiple comparisons (Table 5).

#### 3.2.2. Haplotype Analysis

An assessment of linkage disequilibrium (LD) and haplotype frequencies was carried out for the MMP-9 and TIMP-2 gene pairs with the use of Haploview 4.2 (http://www.broad.mit.edu/mpg/haploview/ (accessed on 20 October 2025)). A marginal LD was established between the analyzed variants for both genes (for MMP-9 rs17576/rs17577 distance 2886 bp, D’ = 0.938, r^2^ = 0.3, and for TIMP-2 rs2277698/rs55743137 distance 168 bp, D’ = 1.0, r^2^ = 0.7). A representational diagram of the LD pattern is shown in Figure 1.

**Figure 1 cimb-48-00025-f001:**
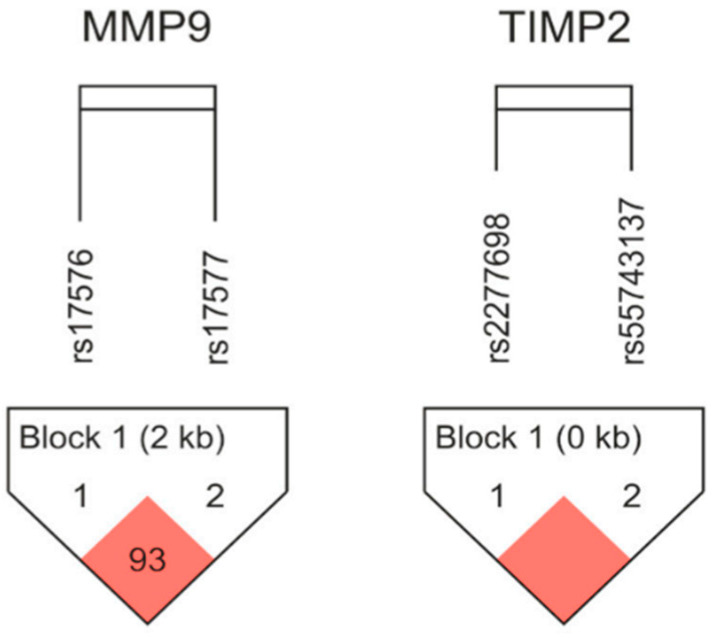
Linkage disequilibrium (LD) plots with two MMP-9 and two TIMP-2 SNPs. According to Lewontin’s D’, which is calculated using Haploview software(version 4.2), statistically significant relationships between a pair of SNPs are shown by red squares. Haplotype frequencies in the newborn groups under study were analyzed; Table 6 displays the findings.

**Table 6 cimb-48-00025-t006:** Haplotype analysis of MMP-9 and TIMP-2 variants in BPD and no-BPD infants.

Genes and SNPs	Haplotype	Frequency (Overall)	Frequency BPD, NO-BPD	Χ^2^	*p*-Value (Person)
MMP9 rs17576/rs17577	AG	0.644	0.617, 0.660	0.377	0.540
	GG	0.201	0.238, 0.179	1.033	0.309
	GA	0.149	0.130, 0.160	0.327	0.567
TIMP2 rs2277698/rs55743137	CT	0.850	0.842, 0.855	0.060	0.807
	TG	0.110	0.118, 0.105	0.089	0.766
	CG	0.040	0.039, 0.040	0.001	0.976

The MMP-9 GG haplotype (rs17576/rs17577) showed the strongest relationship, with a higher prevalence in the BPD group (0.238 vs. 0.179 in no-BPD, χ^2^ = 1.033, *p* = 0.309). There were no statistically significant differences in haplotype distribution observed between the groups for either MMP-9 or TIMP-2 (all *p* > 0.05). Frequencies of haplotypes and LD patterns were analyzed in silico using LDLink (https://ldlink.nih.gov/) based on European populations from the 1000 Genomes Project [18]. The outcome for these populations was comparable to ours. For MMP-9 rs17576/rs17577, D’ = 0.991 and r^2^ = 0.3. The haplotypes were GG = 0.207, GA = 0.174, AG = 0.618, and AA = 0.001. For TIMP-2 rs2277698/rs55743137, D’ = 1.0, r^2^ = 0.6, and the haplotypes were CT = 0.800, TG = 0.125, CG = 0.075, and TT = 0.000.

## 4. Discussion

Prematurity continues to be a major global health issue, with approximately 13.4 million infants born preterm each year [26]. One of the major diseases affecting preterm infants is BPD. This chronic lung disorder can result in long-term respiratory complications and developmental issues. The incidence of BPD varies across countries and populations, which may be attributed to the genetic diversity among said populations [10,27,28].

In our study, we confirmed that primary risk factors such as GA and BW were inversely correlated with the development of BPD [5]. Additionally, we detected statistically significant associations between BPD and a diminished median APGAR score at both the first and fifth minutes after birth, as well as prolonged mechanical ventilation. Further analysis of prematurity-related complications revealed an increased occurrence of IVH, NEC, and ROP with infants diagnosed with BPD. More specifically, findings using the same patient cohort and gene variants suggest that the MMP-1 rs1799750 and TIMP-1 rs4898 variants and their interactions may impact the development of ROP [29]. These specific genes did not present an increased risk for BPD in our analysis. Investigating this apparent difference in correlation will require the use of a larger cohort. Nevertheless, the pathogenesis of BPD remains to be completely understood. It is unclear why some infants experience milder disease or recover without long-term complications, while others develop persistent or severe lung injury. Identifying the genetic factors that predispose infants to BPD could help uncover underlying mechanisms of the disease and potentially predict the severity of the disease and treatment outcomes based on individual genetic profiles.

This study explored the associations of variants in genes encoding MMP-1, MMP-9, TIMP-1, and TIMP-2 with BPD in preterm infants. It was previously reported that the analyzed variants had functional consequences. The rs179975 variant of MMP-1 is an insertion/deletion polymorphism of a single guanine (2G or 1G) found at nucleotide 1707 within the MMP-1 promoter region. The existence of the extra guanine (2G allele) was shown to elevate transcriptional activity, modifying the expression levels of MMP-1 [19,30]. The rs17576 variant of MMP-9 is found within the gelatinase-specific fibronectin type II domain and might improve substrate binding, while rs17577 is situated in the hemopexin domain, which could affect both substrate and inhibitor interactions [20]. Both MMP-9 variants are located in coding regions and alter the amino acid sequence (Gln279Arg and Arg668Gln, respectively). The TIMP-1 rs4898 variant is located in exon 5 on the X chromosome; thus, males are monoallelic (T or C), while females can be TT, CC, or TC. The T allele of the 372 T > C polymorphism has been associated with increased TIMP-1 production [31]. The TIMP-2 rs2277698 polymorphism, located in exon 3, involves a cytosine-to-thymine substitution that does not alter the amino acid sequence (Ser101=), while rs55743137, located in intron 2, involves a thymine-to-guanine change. Demonstrating the importance of these variants in BDP could lead to a better understanding of the disease’s mechanisms.

In our analysis, the single-nucleotide polymorphisms (SNPs) tested did not show statistically significant associations with BPD after correction for multiple testing. Although a nominal association was observed for the MMP1 rs17999750 variant in the dominant model, this did not remain significant after adjustment. Using the adjusted *p*-value for allele frequency analysis, a statistically significant difference was observed for MMP1 rs17999750 (*p* = 0.038). Non-significant trends were observed for both allelic variants of MMP1 rs1799750. Haplotype analysis for MMP-9 and TIMP-2 also did not reveal any statistically significant associations with BPD.

An LD analysis showed a strong correlation between the MMP-9 variants rs17576 and rs17577, as well as between the TIMP-2 variants rs2277698 and rs55743137, in our population. According to the silico results conducted on data from the 1000 Genome project, our results present similar results to other European populations. Haplotype analysis of MMP-GG presented with a slightly increased frequency of BPD cases.

MMPs and TIMPs act as mediators of inflammatory processes and tissue remodeling by interacting with specific extracellular targets, such as components of the extracellular matrix, cytokines, and growth factors [11,12]. It has been suggested that the MMP/TIMP balance plays an important regulatory role in lung development and injury pathogenesis. Fukunaga et al. reported that preterm infants who developed BPD had elevated MMP-9/TIMP-1 ratios in cord blood, which was correlated with the requirement for prolonged oxygen therapy [14]. Further, Lee et al. presented that low TIMP-2 serum levels at birth could serve as a predictor for the subsequent development of BPD, potentially by contributing to early lung inflammation [15]. Our finding that the TIMP-2 rs2277698 TT genotype is a risk factor for BPD aligns with the crucial role of TIMP-2 in regulating lung ECM homeostasis. A relative deficiency in functional TIMP-2, potentially influenced by this genetic variant, could lead to unopposed MMP activity, aberrant alveolarization, and impaired vascular development, which are hallmarks of BPD pathology. The trend for a protective effect of the MMP-9 rs17576 AG genotype in the overdominant model may suggest a complex role for this protease, where heterozygosity confers a more balanced proteolytic environment compared to either homozygous state.

Among other factors that may modulate the influence of the studied genetic variants, postnatal infections and inflammation can be mentioned, which have been identified as significant risk factors for BPD [10]. In this study, we also observed that late-onset infection was significantly more frequent in the BPD group (31.6% vs. 9.7%, *p* = 0.007). The interaction between genetic factors in the MMP/TIMP pathways and postnatal inflammation likely plays a critical role in lung injury development and subsequent healing in preterm infants.

We are slowly learning that the genetic background of BPD in Polish infants is more complicated than we thought. Some studies point to specific genes, like certain variants of IL-1 and IL-6, being seen more often in babies with BPD [32]. Single-nucleotide vitamin D receptor polymorphisms, specifically allele C of apal, more than doubled the risk of BPD [33]. In contrast, some studies showed null results, such as a lack of association between fibronectin-1 gene polymorphisms and BPD [34]. These studies highlight that the genetic profile of BPD is highly nuanced. Further investigation into these relationships is crucial to help establish strategies towards future treatments.

The main strength of this study is its ability to focus on a concise and defined cohort of Caucasian neonates. This heavily enhanced the reliability of the data and opened the potential for inclusion in future meta-analyses. That being said, the primary limitation within this study is its relatively small sample size, which should be validated with the use of a larger cohort. This smaller scale likely contributed to our inability to detect significant associations for specific MMP-9 and TIMP-2 gene variants.

## 5. Conclusions

In conclusion, our findings suggest a potential association between the MMP1 rs17999750 allele and BPD in premature infants. However, no statistically significant evidence was found for the individual role of MMP1 rs1799750, MMP-9 rs17576 and rs17577, TIMP-2 rs2277698, or TIMP-1 rs4898 in the development of this disease.

## Figures and Tables

**Table 1 cimb-48-00025-t001:** Characteristics of selected genes and polymorphisms.

Gene	rs Number	Position (GRCh38.p14)	Allele	Variant Type	MAF
MMP-1	rs1799750	chr11:102799765–102799766	delG	Promoter	delG = 0.4960
MMP-9	rs17576	chr20:46011586	A > G	Coding Gln279Arg	G = 0.3807
MMP-9	rs17577	chr20:46014472	G > A	Coding Arg668Gln	A = 0.1750
TIMP-1	rs4898	chrX:47585586	T > C	Coding Phe124=	C = 0.4650
TIMP-2	rs2277698	chr17:78870935	C > T	Coding Ser101=	T = 0.1252
TIMP-2	rs55743137	chr17:78871103	G > T	Intronic	G = 0.1998

Abbreviations: GRCh38.p14—Genome Reference Consortium Human Build 38 patch release 14. MAF—minor allele frequency based on data from the 1000 Genomes Project for the European population.

**Table 2 cimb-48-00025-t002:** Primers and PCR-RFLP conditions for studied genetic variants (sequence of primers according to [18,19,20,21,22]).

Gene and Variant	Sequence of Primers	Temperature of Primer Attachment	Restriction Enzyme	PCR Products [bp]
MMP-1 rs179975	5’-TGA CTT TTA AAA CAT AGT CTA TGT TCA-3’ 5-TCT TGG ATT GAT TTG AGA TAA GTC ATAGC-3’	50 °C	AluI	1G 241, 28 2G 269
MMP-9 rs17576	5’-GAGAGATGGGATGAACTG-3’ 5’-GTGGTGGAAATGTGGTGT-3’	60 °C	MspI (HpaII)	A 252, 187 G 187, 129, 123
MMP-9 rs17577	5’-ACA CGC ACG ACG TCT TCC AGT ATC-3’ 5-GGG GCA TTT GTT TCC ATT TCC A-3’	63 °C	TaqI	G 115, 23 A 138
TIMP-1 rs4898	5’-GCA CAT CAC TAC CTG CAG TCT-3 5’-GAA ACA AGC CCA CGA TTT AG-3’	54 °C	BauI (BssI)	T 175 C 153, 22
TIMP-2 rs2277698	5’-CCA GGA AAT TGG CAG GTA GT-3’ 5’-GAA TTC ACC AAC TGT GTG GC-3’	60 °C	BsrI	C 369 T 231, 138
TIMP-2 rs55743137	5’-CCT TTG AAC ATC TGG AAA GAC AA-3’ 5’-TAA CCC ATG TAT TTG CAC TTC CT-3’	58 °C	AluI	T 160 G 108, 52

**Table 3 cimb-48-00025-t003:** Characteristics of patients.

Characteristic	I No-BPD (N = 62)	II BPD (N = 38)	II vs. I *p*-Value
Sex, *n* (%) Female Male	31 (50.0) 31 (50.0)	16 (42.1) 22 (57.9)	0.438 ^a^
Gestational age (weeks), median (range)	30 (24–33)	26 (22–31)	<0.0001 ^d^
Birth weight (grams), median (range)	1350 (535–2010)	883 (432–1700)	<0.0001 ^c^
Apgar score, median (range) 1st minute 5th minute	6 (1–10) 8 (3–10)	4 (1–9) 7 (1–10)	<0.0001 ^d^ <0.0001 ^d^
Birth asphyxia, *n* (%)	4 (6.5)	5 (13.2)	0.293 ^b^
Mechanical ventilation (days), median (range)	7 (1–33)	43 (1–146)	<0.0001 ^d^
Intrauterine infection, *n* (%)	33 (53.2)	25 (65.8)	0.212 ^a^
Late-onset infection, *n* (%)	6 (9.7)	12 (31.6)	0.007 ^b^
IVH, *n* (%)	10 (16.1)	19 (50.0)	0.0003 ^a^
NEC, *n* (%)	16 (25.8)	19 (50.0)	0.014 ^a^
ROP (any stage), *n* (%)	21 (33.9)	29 (76.3)	<0.0001 ^a^

Abbreviations: BPD (Bronchopulmonary Dysplasia), IVH (Intraventricular Hemorrhage), NEC (Necrotizing Enterocolitis), and ROP (Retinopathy of Prematurity); statistical analysis: ^a^—χ^2^ test, ^b^—χ^2^ test with Yate’s correction, ^c^—t-student test, and ^d^—Mann–Whitney test.

**Table 4 cimb-48-00025-t004:** Distribution of studied variants in BPD and no-BPD subjects with analysis of differences in allele frequency.

SNP	Allele	NO-BPD (N = 124)	BPD (N = 76)	Crude OR (95% CI)	*p*-Value	Adjusted AOR (95% CI)	Adj. *p*-Value
MMP1 rs1799750	1G	65 (52.42%)	46 (60.53%)	0.72 (0.40–1.28)	0.263	0.40 (0.16–0.95)	0.038
	2G	59 (47.58%)	30 (39.47%)				
MMP9 rs17576	A	82 (66.13%)	48 (63.16%)	1.14 (0.63–2.07)	0.669	1.04 (0.43–2.49)	0.938
	G	42 (33.87%)	28 (36.84%)				
MMP9 rs17577	G	104 (83.87%)	65 (85.53%)	0.88 (0.40–1.96)	0.754	1.18 (0.36–3.86)	0.785
	A	20 (16.13%)	11 (14.47%)				
TIMP1 rs4898	Female newborns						
	T	34 (54.84%)	17 (53.12%)	1.07 (0.46–2.52)	0.874	2.11 (0.50–8.88)	0.310
	C	28 (45.16%)	15 (46.88%)				
	Male newborns						
	T	28 (45.16%)	22 (50.00%)	0.82 (0.38–1.79)	0.623	0.64 (0.20–2.07)	0.459
	C	34 (54.84%)	22 (50.00%)				
TIMP2 rs2277698	C	111 (89.52%)	67 (88.16%)	1.15 (0.47–2.83)	0.766	0.63 (0.16–2.47)	0.502
	T	13 (10.48%)	9 (11.84%)				
TIMP2 rs55743137	T	106 (85.48%)	64 (84.21%)	1.10 (0.50–2.44)	0.807	1.00 (0.31–3.21)	0.998
	G	18 (14.52%)	12 (15.79%)				

*p* adj.—adjusted for newborn’s weight, week of pregnancy, late-onset infection and time of ventilation.

**Table 5 cimb-48-00025-t005:** Genotype distribution in the studied infants and analysis of the association between individual variants of MMP-1, MMP-9, TIMP-1, and TIMP-2 genes and the occurrence of BPD.

Gene. SNP	Genotypes and Tested Models	I NO-BPD N (%)	II BPD N (%)	II vs. I					
				Crude			Adjusted		
				OR (95%CI)	*p*	AIC	AOR (95%CI)	*p* adj.	AIC
MMP1 rs1799750	1G/1G	18 (29.0)	16 (42.1)	1.00	0.405	137.0	1.00	0.063	84.1
	1G/2G	29 (46.8)	14 (36.8)	0.54 (0.21–1.37)			0.22 (0.05–0.93)		
	2G/2G	15 (24.2)	8 (21.1)	0.60 (0.20–1.79)			0.20 (0.03–1.23)		
	Dominant	44 (71.0)	22 (57.9)	0.56 (0.24–1.31)	0.183	135.0	0.22 (0.06–0.84)	0.019	82.1
	Recessive	47 (75.8)	30 (78.9)	0.84 (0.32–2.21)	0.716	136.7	0.50 (0.11–2.20)	0.347	86.7
	Overdominant	33 (53.2)	24 (63.2)	0.66 (0.29–1.52)	0.329	135.9	0.41 (0.12–1.39)	0.141	85.4
	Log-additive	62 (62.0)	38 (38.0)	0.75 (0.43–1.29)	0.290	135.7	0.41 (0.17–1.00)	0.038	83.3
MMP9 rs17576	AA	26 (41.9)	18 (47.4)	1.00	0.142	134.9	1.00	0.110	85.2
	AG	30 (48.4)	12 (31.6)	0.58 (0.23–1.42)			0.39 (0.10–1.43)		
	GG	6 (9.7)	8 (21.1)	1.93 (0.57–6.51)			2.94 (0.41–20.96)		
	Dominant	36 (58.1)	20 (52.6)	0.8 (0.36–1.81)	0.596	136.5	0.62 (0.19–1.96)	0.411	86.9
	Recessive	56 (90.3)	30 (78.9)	2.49 (0.79–7.84)	0.117	134.4	4.27 (0.64–28.32)	0.130	85.3
	Overdominant	32 (51.6)	26 (68.4)	0.49 (0.21–1.15)	0.096	134	0.33 (0.09–1.15)	0.071	84.4
	Log-additive	62 (62.0)	38 (38.0)	1.13 (0.63–2.01)	0.681	136.6	1.03 (0.44–2.44)	0.939	87.6
MMP9 rs17577	GG	44 (71.0)	27 (71.1)	1.00	0.687	136.8	1.00	0.812	89.2
	GA	16 (25.8)	11 (28.9)	1.12 (0.45–2.77)			1.34 (0.36–5.07)		
	AA	2 (3.2)	0 (0.0)	—			—		
	Dominant	18 (29.0)	11 (28.9)	1.00 (0.41–2.43)	0.993	136.8	1.29 (0.34–4.81)	0.708	87.5
	Recessive	60 (96.8)	38 (100)	—	0.524	134.9	—	0.633	87.4
	Overdominant	46 (74.2)	27 (71.1)	1.17 (0.47–2.89)	0.732	136.7	1.36 (0.36–5.16)	0.651	87.4
	Log-additive	62 (62.0)	38 (38.0)	0.88 (0.39–1.97)	0.687	136.7	1.20 (0.35–4.12)	0.776	87.5
TIMP1 rs4898	Female newborns								
	TT	7 (22.6)	3 (18.8)	1.00	0.949	66.2	1.00	0.311	37.6
	TC	20 (64.5)	11 (68.8)	1.28 (0.28–5.98)			5.29 (0.23–120.23)		
	CC	4 (12.9)	2 (12.5)	1.17 (0.13–10.22)			31.61 (0.23–)		
	Dominant	24 (77.4)	13 (81.2)	1.26 (0.28–5.73)	0.759	64.2	5.93 (0.29–122.54)	0.227	36.5
	Recessive	27 (87.1)	14 (87.5)	0.96 (0.16–5.93)	0.969	64.3	7.03 (0.18–271.05)	0.281	36.8
	Overdominant	11 (35.5)	5 (31.2)	1.21 (0.33–4.39)	0.771	64.2	1.43 (0.13–15.71)	0.771	37.8
	Log-additive	31 (66)	16 (34)	1.11 (0.39–3.16)	0.847	64.2	5.56 (0.50–61.46)	0.127	35.6
	Male newborns								
	TT	14 (45.2)	11 (50.0)	1.00	0.728	75.8	1.00	0.5984	53.7
	TC	—	—	—			—		
	CC	17 (54.8)	11 (50.0)	0.82 (0.28.2.46)			1.55 (0.30–8.08)		
	Log-additive	31 (58.5)	22 (41.5)	1.21 (0.41–3.63)		75.8	1.55 (0.30–8.08)		53.7
TIMP2 rs55743137	TT	45 (72.6)	28 (73.7)	1.00	0.539	137.6	1.00	0.705	88.9
	TG	16 (25.8)	8 (21.1)	0.80 (0.30–2.12)			0.73 (0.18–2.93)		
	GG	1 (1.6)	2 (5.3)	3.21 (0.28–37.11)			3.98 (0.07–225.45)		
	Dominant	17 (27.4)	10 (26.3)	0.95 (0.38–2.35)	0.904	136.8	0.85 (0.23–3.19)	0.815	87.6
	Recessive	61 (98.4)	36 (94.7)	3.39 (0.3–38.71)	0.308	135.8	4.23 (0.07–240.08)	0.479	87.1
	Overdominant	46 (74.2)	30 (78.9)	0.77 (0.29–2.01)	0.587	136.5	0.71 (0.18–2.82)	0.623	87.4
	Log-additive	62 (62.0)	38 (38.0)	1.10 (0.51–2.37)	0.813	136.8	1.00 (0.32–3.16)	0.998	87.6
TIMP2 rs2277698	CC	49 (79.0)	31 (81.6)	1.00	0.112	134.1	1.00	0.228	86.7
	CT	13 (21.0)	5 (13.2)	0.61 (0.2–1.87)			0.32 (0.06–1.72)		
	TT	0 (0.0)	2 (5.3)	—			—		
	Dominant	13 (21.0)	7 (18.4)	0.85 (0.31–2.37)	0.756	136.7	0.43 (0.09–2.08)	0.285	86.5
	Recessive	62 (100.0)	36 (94.7)	—	0.142	132.9	—	0.302	86.5
	Overdominant	49 (79.0)	33 (86.8)	0.57 (0.19–1.75)	0.315	135.8	0.30 (0.05–1.63)	0.149	85.5
	Log-additive	62 (62.0)	38 (38.0)	1.14(0.48–2.7)	0.112	136.7	0.65 (0.18–2.39)	0.515	87.2

*p* adj.—adjusted for newborn’s weight, week of pregnancy, late-onset infection and time of ventilation.

## Data Availability

The data that support the findings of this study are available from the corresponding author upon reasonable request.

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
