# Peer review of "Analysis of the Role of Gene Variants in Matrix Metalloproteinases and Their Tissue Inhibitors in Bronchopulmonary Dysplasia (BPD): A Study in the Polish Population"

_cimb, 2025, doi:10.3390/cimb48010025_

Round 1
Reviewer 1 Report
Comments and Suggestions for Authors
This manuscript investigates the association between genetic variants of MMPs (MMP-1, MMP-9) and TIMPs (TIMP-1, TIMP-2) with bronchopulmonary dysplasia (BPD) in a cohort of Polish preterm infants. The study is well‐structured, the methodology is generally sound, and the results contribute incremental evidence in a biologically plausible research area. However, the manuscript has several limitations—including insufficient adjustment for confounders, and limited mechanistic interpretation—that need to be addressed before acceptance.
- Only gestational age and birth weight were included in the adjusted logistic models. Important clinical covariates—such as infection, ventilation duration, maternal factors (smoking, preeclampsia)—were shown to differ between groups and should be considered in multivariable analyses. Without this, the associations may be confounded.
- Power calculations should be added, or the authors should discuss the potential for false-positive or false-negative results more explicitly.
- The manuscript includes numerous statistical tests across variants and inheritance models. However, no Bonferroni/FDR correction is reported. This raises concerns regarding false-positive significance.
- Although prior literature is cited, the discussion should better integrate the present findings into established biological pathways
- Several references contain inconsistent formatting, broken links, or non-standard citation structures.
Author Response
Only gestational age and birth weight were included in the adjusted logistic models. Important clinical covariates—such as infection, ventilation duration, maternal factors (smoking, preeclampsia)—were shown to differ between groups and should be considered in multivariable analyses. Without this, the associations may be confounded.
We updated the regression model with infection and time of ventilation as you recommended (Tables 4 and 5). None of the mothers had preeclampsia or were smokers. We add on this information in the exclusion criteria for the study group in section 2.1.
Power calculations should be added, or the authors should discuss the potential for false-positive or false-negative results more explicitly.
From the introduction: “One study on a Polish population found that 52.7% of infants born before 32 weeks and weighing ≤1500g developed BPD [8]”.
Added in part 2.5
We computed the statistical power under the assumption that 52.7% of preterm infants in the Polish research had BPD [8]. In our cohort, 38% of preterm infants had BPD. With a sample size of N=100 and alpha=0.05, the power of the test was 84.4%.
The manuscript includes numerous statistical tests across variants and inheritance models. However, no Bonferroni/FDR correction is reported. This raises concerns regarding false-positive significance.
Added in part 2.5
P-values less than 0.008 were considered statistically significant after applying a Bonferroni correction for the six SNPs analysed.
Added in part 3.2.1
None of the SNPs tested is significant after the Bonferroni correction (p<0.008) for multiple testing
Although prior literature is cited, the discussion should better integrate the present findings into established biological pathways
This comment is very vague and does not specify much but nonetheless i still tried to add a more clear limitation and reword it.
Several references contain inconsistent formatting, broken links, or non-standard citation structures.
In regards to this, I used the mdpi zotero option to maintain a consistent standard and personally reinputted every single citation in the paper
Reviewer 2 Report
Comments and Suggestions for Authors
This original Study investigated examine the relationship between variants of MMP-1, MMP-9, TIMP-1, and TIMP-2 genes and BPD in a premature infants. This is relative new but frequent syndrome of mutigen origin. The Study is well design, in a combitaion of clinical. and genetic factors. It is also well weitten and present in tables and Figures. This pathology is ussulally linked to connective tissue danage. As a result, TIMP-2 rs2277698 variant may impact the development of BPD in a Polish cohort of premature infants. This is the normal than 1 from 5 show positive correlation in disease, so I support the results. Also it is well English style.
The re are some ussues that could be corrected:
1) You need not to repeat everywhere about " Polish cohort" - this actually restricts your results. In title and Methods is enough.
2) The abstract is not structured, please and aim, Methods and Results (conclusions already existing).
3) Please add limitations at the end.
4) Did you estimated the power of the Study? The Number of patients is low.
Author Response
You need not to repeat everywhere about " Polish cohort" - this actually restricts your results. In title and Methods is enough.
Thank you for pointing this out, i went through and significantly lowered the use of “polish cohort”
The abstract is not structured, please and aim, Methods and Results (conclusions already existing).
I went over the abstract and presented it as requested
Please add limitations at the end.
A limitation has been added in the abstract and end of discussion
Did you estimated the power of the Study? The Number of patients is low.
From the introduction: “One study on a Polish population found that 52.7% of infants born before 32 weeks and weighing ≤1500g developed BPD [8]”.
Added in part 2.5
We computed the statistical power under the assumption that 52.7% of preterm infants in the Polish research had BPD [8]. In our cohort, 38% of preterm infants had BPD. With a sample size of N=100 and alpha=0.05, the power of the test was 84.4%.